# Biochemical, Micronutrient and Physicochemical Properties of the Dried Red Seaweeds *Gracilaria edulis* and *Gracilaria corticata*

**DOI:** 10.3390/molecules24122225

**Published:** 2019-06-14

**Authors:** Thomas Rosemary, Abimannan Arulkumar, Sadayan Paramasivam, Alicia Mondragon-Portocarrero, Jose Manuel Miranda

**Affiliations:** 1Department of Oceanography and Coastal Area Studies, School of Marine Sciences, Alagappa University, Thondi Campus, Thondi-623 409, India; rosemarythomas@gmail.com (T.R.); aruul3@gmail.com (A.A.); 2Department of Biotechnology, Achariya Arts and Science College, Villianur, Puducherry-60 110, India; 3Departamento de Química Analítica, Nutrición y Bromatología, Facultade de Veterinaria, Universidade de Santiago de Compostela, Pabellón 4, Planta Baja, 27002 Lugo, Spain; aliciamondragon@yahoo.com (A.M.-P.); josemanuel.miranda@usc.es (J.M.M.)

**Keywords:** water holding capacity, seaweeds, *Gracilaria*, carbohydrates, fatty acids, vitamins

## Abstract

The present study sought to evaluate the nutritional composition and physicochemical properties of two dried commercially interesting edible red seaweeds, *Gracilaria corticata* and *G. edulis*. Proximate composition of the dried seaweeds revealed a higher content in carbohydrates (8.30 g/100 g), total crude protein (22.84 g/100 g) and lipid content (7.07 g/100 g) in *G. corticata* than in *G. edulis*. Fatty acids profile showed that *G. corticata* samples contain higher concentrations of saturated fatty acids, such as palmitic and stearic acids, and polyunsaturated ones such as α-linolenic and docosahexaenoic acids. Contrariwise, *G. edulis* contained higher amounts of monounsaturated oleic acid. Total amino acid content was 76.60 mg/g in *G. corticata* and 65.42 mg/g in *G. edulis*, being the essential amino acid content higher in *G. edulis* (35.55 mg/g) than in *G. corticata* (22.76 mg/g). Chlorophyll *a* was found in significantly higher amounts in *G. edulis* (17.14 μg/g) than *G. corticata*, whereas carotenoid content was significantly higher in *G. corticata* (12.98 μg/g) than in *G. edulis*. With respect to physical properties, both water- and oil-holding capacities were similar in both seaweeds, whereas swelling capacity was higher in *G. edulis*. In view of the results, the present study suggests that *G. corticata* and *G. edulis* contains important nutrients for human health and are possible natural functional foods.

## 1. Introduction

Seaweeds are very important natural resources from the oceans that are employed as human foods and animal feeds in their whole form, and as sources of polysaccharides (mainly alginates, carrageenans and agar), carotenoids, lipids, vitamins, minerals, dietary fiber, proline and amino acids for use in food and pharmaceutical industry [1]. Seaweeds have been included for a long time in the traditional diet of East Asian countries such as Japan, Korea and China; more recently, their presence in all forms in the diet of Western countries has been progressively increasing [2].

Seaweeds are considered healthy foods because, despite their low caloric content, they are rich in important nutrients such as protein, essential amino acids, vitamins, minerals and some bioactive compounds [1]. Seaweeds are also an excellent source of both soluble and insoluble dietary fiber. Among red algae, the genus *Gracilaria* contains a broad diversity of valuable contents for human nutrition and are one of the world’s most cultivated and valuable marine seaweed [3]. Its lipid content is low (1–5% dry weight, DW) [1], but it contains docosahexaenoic acid (DHA) which is recognized as the most important *n*-3 polyunsaturated fatty acid (PUFA) to reduce the risk of cardiovascular diseases [4,5]. In particular, *n*-3 PUFAs act as excellent antioxidants, strengthening the cell membrane, repairing damaged cells and tissues, improving heart function and fighting against cancer [6]. *n*-3 PUFAs were also found to prevent the growth of atherosclerotic plaque that affects blood clotting and blood pressure and improve the immune function, while *n*-6 PUFAs decrease low-density lipoprotein cholesterol and may also decrease high-density lipoprotein, cholesterol which reduces heart disease risk [7].

With respect to their protein content, the most abundant amino acids in *Gracilaria* species are aspartic acid, alanine, glutamic acid and glutamine. These amino acids provide the typical flavor of algae and accumulate in response to stress conditions [8]. *Gracilaria* is also a good source of both soluble and insoluble dietary fiber, so it can be employed as a potential alternative to cereal-based fiber in Western countries [1]. Soluble dietary fiber helps to increase viscosity and reduce glycemic response and plasma cholesterol in humans [1]. Insoluble dietary fiber improves the bulking effect caused by water absorption in feces and thus contributes to weight management, improvement of cardiovascular and gastrointestinal functions and cancer prevention [1]. Polysaccharides isolated from red seaweeds show potent antibacterial, antiviral, antioxidant, anticoagulant and anti-inflammatory activity [9].

Seaweeds such as *Gracilaria* can concentrate minerals from seawater and reach a mineral content 10–20 times higher than that of terrestrial plants [10]. Consequently, they are a valuable source of minerals, with important human nutrition functions [11,12]. Chlorophyll, an important pigment constituent present in algae, has positive effects on inflammation, oxidation and wound healing [13]. Chlorophyll acts directly as a reducer of free radicals and has the potential to protect lymphocytes against oxidative DNA damage by free radicals [14]. Moreover, a large number of potentially bioactive compounds such as phenols, polyphenols, terpenes, steroids, halogenated ketones and alkanes, fucoxanthin, polyphloroglucinol and bromophenols have been isolated [15,16,17].

However, the nutrient profile of seaweeds such as *Gracilaria* is influenced by different factors such as seaweed species, habitat, maturity stage, season, water temperature and the sampling conditions and method employed in the determinations [1,2]. *Gracilaria edulis* and *G. corticata* is abundantly available in almost all seasons in Palk Bay, on the southeast coast of India, rather than other *Gracilaria* sp. Both *G. edulis* and *G. corticata* are commercially important and commonly edible seaweeds in India. These two algae exhibited in a previous work high biological activities (proximate composition, antioxidant, antibacterial, and biopreservative effects in seafoods during preservation and extended shelf life than other *Gracilaria* species in a previous work [18]. Thus, the present study sought to evaluate and compare the chemical composition (proximate composition, lipid profile, amino acids, vitamins and pigments such as chlorophyll and carotenoids) and physicochemical properties of both, *Gracilaria corticata* and *Gracilaria edulis* from the Thondi coast of Palk Bay, southeast India.

## 2. Results

Proximate, polysaccharide content and fatty acids profile of both *G. corticata* and *G. edulis* in a DW basis are shown in Table 1.

The crude polysaccharide content found for *G. corticata* and *G. edulis* was 49.64 g/100 g and 38.02 g/100 g, respectively. The moisture content (in dried seaweeds) of *G. corticata* and *G. edulis* was 8.40 g/100 g and 10.40 g/100 g, respectively. With respect to proximate composition, important differences were obtained for the two seaweeds investigated. Carbohydrates and fat content were significantly higher in *G. corticata*, whereas protein content was significantly higher in the case of *G. edulis.*

With respect to fatty acids profile, total fatty acid content, expressed as g fatty acids methyl esters (FAME)/100 g total fat, of *G. corticata* and *G. edulis* was 5.49 ± 0.30 g/100 g and 3.92 ± 0.13 g/100 g, respectively (Table 2). The main saturated fatty acids (SFAs) found in both *G. corticata* and *G. edulis* were palmitic acid (C16:0), margaric acid (C17:0) and stearic acid (C18:0). With respect to PUFAs, linoleic acid (C18:2*n*-6), α-linolenic acid (C18:3*n*-3), stearidonic acid (C18:4*n*-3) and DHA (C22:6*n*-3) were found in both seaweeds. In the case of monounsaturated fatty acids (MUFAs), only oleic acid (C18:1) was detected in relevant amounts in both *G. corticata* and *G. edulis*. Margaric, linoleic and stearidonic acids were found in similar amounts in *G. corticata* and *G. edulis*. Palmitic, stearic, α-linolenic acid were found in higher amounts in *G. corticata* than in *G. edulis*. Overall, in G. *corticata*, SFAs accounted 49.4% of total fatty acids, MUFAs accounted a 3.3% and PUFAs accounted a 47.3%, whereas in the case of *G. edulis*, SFAs accounted 43.9% of total fatty acids, MUFAs accounted a 27% of total fatty acids, and PUFAs accounted a 29%. 

The protein of *G. corticata* and *G. edulis* is shown in Table 3. The total amino acid content was higher in *G. corticata* (76.60 ± 5.14 mg/g), than in *G. edulis* (65.42 ± 3.58 mg/g). These values are comparable to their corresponding crude protein content of 22.84 ± 0.87 and 25.29 ± 0.67 g/100 g, respectively, indicating that the amount of non-protein nitrogenous materials in these red seaweeds is low. 

Nine essential amino acids (EAAs), and 11 non-essential amino acids (NEAAs), were found in both *G. corticata* and *G. edulis*. Total EAAs where significantly higher in *G. edulis* (35.55 ± 1.75 mg/g) than in *G. corticata* (22.76 ± 1.81 mg/g), whereas total NEAAs where higher in *G. corticata* (36.14 ± 3.33 mg/g) than in *G. edulis* (29.86 ± 1.83 mg/g). The EAAs/total amino acid ratio suggests that more than 50% of the amino acids found in *G. edulis* are EAAs. The results also indicate a good ratio of essential amino acids to non-essential amino acids in *G. corticata* (0.62 ± 0.54 mg/g) and *G. edulis* (1.19 ± 0.95 mg/g). It was noted that a much higher concentration of the essential amino acid threonine (20.57 ± 0.62 mg/g) was found in *G. edulis* than in *G. corticata*. Contrariwise, alanine content was much higher in *G. corticata* (21.11 ± 0.54 mg/g) than in *G. edulis* (1.46 ± 0.18 mg/g). Aspartic acid content was similar in both seaweeds.

The mineral content of *G. corticata* and *G. edulis* is shown in Table 4. *G. corticata* showed a higher content of Mg (463.23 ± 8.87 mg/kg) and Fe (1072.48 ± 20.97 mg/kg) than *G. edulis*. Moreover, *G. edulis* was found to possess more of trace elements like Zn (42.73 ± 2.12 mg/kg) and Cu (14.61 ± 0.46 mg/kg) than *G. corticata.* In view of the present results, both *G. corticata* and *G. edulis* contain an adequate amount of minerals, which suggests that these seaweeds could act as important sources of mineral supplements which are essential for human nutrition.

For both *G. corticata* and *G. edulis,* the presence of water-soluble vitamins (vitamin B_1_, vitamin B_2_, vitamin B_3_, vitamin B_6_, vitamin B_9_ and vitamin C) and fat-soluble vitamins (vitamin A and vitamin E) was found, as shown in Table 5. *G. corticata* had a higher vitamin A (2.67 ± 0.31 mg/g vs. 2.14 ± 0.17 mg/g) and vitamin B_9_ contents (1.00 ± 0.07 mg/g vs. 0.45 ± 0.06 mg/g) than *G. edulis,* whereas *G. edulis* showed a significantly higher content of vitamin B_2_ (1.54 ± 0.39 mg/g vs. 0.05 ± 0.01 mg/g) and vitamin B_6_ (4.77 ± 0.23 mg/g vs. 3.79 ± 0.30 mg/g) than *G. corticata*. 

The methanolic extracts of *G. corticata* and *G. edulis* at 1 mg/mL concentration indicate the presence of three major compounds, chlorophyll *a*, chlorophyll *b* and carotenoids, in *G. corticata* (Retention factor (Rf) value = 0.97, 0.92 and 0.95, respectively) and *G. edulis* (Rf = 0.96, 0.96 0.84, respectively). *G. corticata* and *G. edulis* contained 8.96 ± 0.39 µg/g and 17.14 ± 0.55 µg/g of chlorophyll *a* and 7.74 ± 0.33 µg/g and 8.44 ± 0.63 µg/g of chlorophyll *b*, respectively. With respect to the carotenoid content, it was higher for *G. corticata* (12.82 ± 0.50 µg/g) than for *G. edulis* (2.99 ± 0.56 µg/g). 

Table 6 shows the swelling capacity (SWC), water-holding capacity (WHC) and oil-holding capacity (OHC) of *G. corticata* and *G. edulis*. In general, as temperature varied, the SWC and WHC of *G. corticata* and *G. edulis* powder varied, due to an increase in the solubility of the dietary fiber and the presence of protein in *G. corticata* and *G. edulis*. However, it also reaches significant differences for the case of SWC, whereas no statistical differences were obtained for WHC or OHC. The SWC of *G. edulis* were higher than *G. corticata* at both 25 °C and 37 °C (8.66 ± 0.53 mL/g vs. 7.90 ± 0.32 mL/g, and 7.70 ± 0.60 mL/g vs. 5.70 ± 0.65 mL/g, respectively). 

With respect to the WHC of *G. corticata* and *G. edulis,* values of 4.03 ± 0.39 and 4.09 ± 0.28 g/g, respectively, were obtained at 25 °C, reduced to 3.96 ± 0.58 g/g in *G. corticata* and 3.64 ± 0.18 g/g in *G. edulis* at 37 °C. In this study, both *G. corticata* and *G. edulis* exhibited similar OHC values (about 2 g/g) at both 25 °C and 37 °C. 

## 3. Discussion

With respect to proximate content, the moisture of *G. corticata* and *G. edulis* was lower than most results obtained for *Gracilaria* sp. in general, such as the 12.15 g/100 g obtained for *G. acerosa* [6] the 19.2 g/100 g for *G. edulis* [8], and the 12.86 g/100 g for *G. edulis* [1], but higher than the 5.32 g/100 g obtained for *G. changii* [19]. In this work, *G. edulis* showed a higher ash content in a DW basis than *G. corticata*. Similarly, it was reported an ash content of 8.70 g/100 g in *G. edulis* [8], whereas other authors reported a higher ash content (40.30 g/100 g) in *G. changii* [19] than those found in the present work. A high ash content shows the presence of appreciable amounts of diverse minerals found in both seaweeds. A similar observation was for *G. changii* [19] in which were found an ash content of 6.40 g/100 g. Interestingly, total dietary fiber is known to have physiological properties for the prevention and treatment of cancer, obesity and diabetes [20,21]. Therefore, *G. corticata* and *G. edulis* may have the potential to be used as a source of dietary fiber in the nutraceutical industries.

Other authors found much lower crude protein contents in *Gracilaria* spp. than those found in the present work. Thus, it was reported a crude protein content of 6.68 g/100 g for *G. edulis* [8], 0.61 g/100 g for *G. acerosa* [6], 12.57 g/100 g in *G. changii* [19] or 19.70 g/100 g for *G. cervicornis* [22]. Moreover, the high protein content of *G. corticata* and *G. edulis* indicates that these seaweeds may be considered as potential marine plant sources of protein [22]. Proteins from seaweeds can have antibacterial, antioxidant, immunostimulating, antithrombotic and anti-inflammatory activities. Consequently, they can be used for prevention and treatment of hypertension, diabetes and hepatitis among other positive effects in the organism [20].

The total carbohydrate content of both *G. corticata* and *G. edulis* was markedly lower than that reported [8] for *G. edulis* (10.2 g/100 g) or the 29.44 g/100 g reported for *G. changii* [19]. However, other authors found lower carbohydrate content in *Gracilaria* species, such as *G. acerosa*, for which was reported a carbohydrate content of 1.05 g/100 g [6]. The wide variation in the carbohydrate content observed in red and brown seaweed species might be due to the influence of different factors like salinity, temperature and sunlight intensity [2]. Moreover, carbohydrate content is also influenced by biomass, which reveals the link between growth and carbohydrate content [23].

In general, seaweeds have a low fat content [23]; that makes seaweeds low-calorie foods and in the present work both seaweeds contained fat amounts of 7.07 g/100 g DW seaweed (*G. corticata)* and 4.71 g/100 g DW seaweed (*G. edulis*). These results are lower than those obtained by other authors [8], whose reported a crude lipid content for *G. edulis* of 8.30 g/100 g but significantly that the 0.3% reported for *G. changi* [19], or the 1.7–3.6% reported for *G. fisheri* and *G. tenuistipitata* [5]. Thus, *Gracilaria* content in fat can widely vary depending on the species and source. 

Polysaccharides are polymers composed of at least 10 monosaccharides linked by glycosidic bonds [9]. Recently, seaweed polysaccharides have been given large attention by the scientific community due to their outstanding bioactivities and correspondingly low toxicity [9]. They have been shown to have other beneficial health effects, including their prebiotic effect and antioxidant or anti-inflammatory activity [20]. The polysaccharide content obtained in the present work was higher than the polysaccharide extracted from *Gracilaria* species in previous works, such as 29.08 g/100 g [24], 27.20 g/100 g [25], 21.40 g/100 g [26], and 32.80 g/100 g [27]. Contrariwise, it was also reported a higher polysaccharide content in *G. debilis* [28], in the range 52–67 g/100 g. A previous work [29] reported that the polysaccharide yield from *Gracilaria* species varies due to seasonal variations, physiochemical factors, environmental conditions and extraction methods. Additionally, the variations in the polysaccharide content of *Gracilaria* can vary depending on atmospheric temperature at the time of extraction [26]. Hence the present study significantly indicates that the crude polysaccharides present in *G. corticata* and *G. edulis* may exert varied biological activity [25].

With respect to the fatty acids composition, those of seaweeds often differ from those of terrestrial plants whereby seaweeds have a higher proportion of PUFAs than terrestrial vegetables. Red seaweeds are particularly rich in SFAs and PUFAs which have nutritional applications that lead to their extensive use in food, feed, cosmetic, biotechnological and pharmaceutical applications [30,31]. Variation in fatty acid content may also be due to the season of collection as well as other abiotic factors such as nutrition, salinity, light and temperature [8,20]. In the present work, total fatty acids were significantly lower than those obtained by other authors [8], who found 11.41 g/100 g in *G. edulis*. According to this work [8], the most abundant fatty acids in both seaweeds were palmitic, stearic and α-linoleic acid acids. The same fatty acids were also found abundant in *G. changii* [20]. However, our results were significantly lower than those obtained in *G. changii* for DHA content, in which DHA were found as the most abundant fatty acids, with a 48.36% of total fatty acids. The results of the present study revealed that both seaweeds are rich in SFAs and especially in PUFAs, which provide important health benefits. With respect to most commonly found n-3 PUFA, eicosapentaenoic acid (EPA) and DHA, it is common that their contents vary dramatically from *Gracilaria* spp. and even into the same species [32]. No EPA presence were found for the seaweeds tested in the present work. The presence of this n-3 fatty acid in *Gracilaria* spp. is inconstant, because it was found in *G. gracilis* [20], but it was not detected in *G. changii* [19] or *G. edulis* [8]. Fatty acids overall profile obtained in this work were significantly different than 57.5% SFAs, 18.3% MUFAs and 18.4% PUFAs reported for *Gracilaria* sp. [3] or the 7.5% SFAs, 38.3% MUFAs and 51.2% PUFAs 18.4% reported for *Gracilaria changii* [19].

The protein composition found in this work for *G. corticata* and *G. edulis* was lower than those found in a previous work [20], which reported an amino acid content of 91.90 mg/g in *G. changii*. The EAAs/total amino acid ratio was higher than those previously reported [6,19,26]. Aspartic acid content, that is important for the organoleptic point of view because it was reported that it is responsible for the special flavor and taste of seaweeds [33], was in similar contents in both seaweeds.

Seaweeds are one of the richest sources of minerals and trace elements, because the cell-wall polysaccharides and proteins of seaweed contain sulfate, anionic carboxyl and phosphate groups which act as binding sites for metal retention [34]. With respect to the mineral content, *G. corticata* showed a higher content of Mg (463.23 mg/kg) and Fe (1072.48 mg/kg) than *G. edulis*. Moreover, *G. edulis* was found to possess more of trace elements like Zn (42.73 mg/kg) and Cu (14.61 mg/kg) than *G. corticata.* Both seaweeds had a higher or similar content of minerals like Zn, Cu, Mg and Fe when compared with the content of *G. acerosa* [6], *G. edulis* [8], *G. fisheri* and *G. tenuistipidatata* [5] or *G. changii* [19], with the exception of Mg in *G. edulis* which were lower than those found for other previous works as *G. changii* [19]. The ability of seaweeds to accumulate metals will depend on a variety of factors such as location, exposure, salinity, temperature, pH, light, nitrogen content, season, plant age, metabolic processes or the affinity of the plant for each element among others [35]. In view of the present results, both *G. corticata* and *G. edulis* contain an adequate amount of minerals, which suggests that these seaweeds could act as important sources of mineral supplements which are essential for human nutrition.

For both *G. corticata* and *G. edulis,* the presence of water-soluble vitamins (vitamin B_1_, vitamin B_2_, vitamin B_3_, vitamin B_6_, vitamin B_9_ and vitamin C) and fat-soluble vitamins (vitamin A and vitamin E) was found. *G. corticata* had a higher vitamin A (2.67 mg/g vs. 2.14 mg/g) and vitamin B_9_ contents (1.00 mg/g vs. 0.45 mg/g) than *G. edulis,* whereas *G. edulis* showed a significantly higher content of vitamin B_2_ (1.54 mg/g vs. 0.05 mg/g) and vitamin B_6_ (4.77 mg/g vs. 3.79 mg/g) than *G. corticata.* With respect to previously published works, the vitamin content reported for *Gracilaria* species is widely different between the different authors [1,6,8]. Perhaps the more remarkable difference in vitamin content is that in the present work both *G. corticata* and *G. edulis* showed a significantly higher vitamin A content (2.67 and 2.07, respectively) than those previously reported for *G. acerosa* [6] or for *G. edulis* [1]. 

Another important difference was found for the case of vitamin C that showed a higher content than those previously reported [6,8] for *G. edulis* or *G. acerosa*, respectively. The variation in vitamin content may be due to some environmental factors such as salinity, atmospheric temperature, seasonality and methods of preservation and processing [6]. 

*G. corticata* and *G. edulis* contained 8.96 µg/g and 17.14 µg/g of chlorophyll *a* and 7.74 µg/g and 8.44 µg/g of chlorophyll *b*, respectively. In a previous work describing the composition of several seaweeds [36] it was reported that chlorophyll *a* and *b* in red seaweeds ranged from 68 to 162 µg/g and from 25 to 46 µg/g, respectively. Specifically, for *Gracilaria* spp., it was reported high chlorophyll *a* of 577.89 µg/g and low chlorophyll *b* of 1.11 µg/g in *G. changii* [19]. 

With respect to carotenoid content, a higher total carotenoid content was reported for *G. changii* than in the present study (74.22 µg/g) [19]. Carotenoids such as *β*-carotene, lutein, zeaxanthin and antheraxanthin have been identified in red seaweed, including *Gracilaria* species. Further, seaweed carotenoids, especially *β*-carotene, are preferred by the market of natural products, because they are a mixture of cis and trans isomers, which may possess anticancer activity [37].

The SWC, WHC and OHC properties of seaweeds are generally related to their content and type of polysaccharides as well as protein which links to the cell wall of polysaccharide [5]. Previous works described that variations in temperature can widely vary physicochemical properties of seaweeds, due to increase in the solubility of the dietary fiber and the presence of protein [5,8]. However, in our work, only were found significant variations in the case of SWC. Previous works [8] reported a SWC of 20 mL/g in *G. edulis*, higher than those found in the present study. Similarly, a SWC at 37 °C of 7.68 mL/g in *G. changii* was reported [19], whereas for *G. acerosa* [6] an SWC at 37 °C of 5 mL/g was reported.

With respect to the WHC of *G. corticata* and *G. edulis,* a similar observation was also made in a previous work [8], that reported a WHC for *G. edulis* of 3.08 g/g. Other authors found better WHC than in the present work for other *Gracilaria* spp., such as *G. fisheri*, for which a WHC of 5.53 g/g was reported [5], and *G. changii* [26], for which WHC values of 6.15 g/g at 24 °C and 9.93 g/g at 37 °C were reported. Both SWC and WHC of seaweeds might be attributed due to different protein content and increases in the number and nature of the water binding sites on the protein molecules [38].

OHC is another functional property of food ingredients used in formulated foods for consumption. Ingredients with high OHC values allow the stabilization of food emulsions and high-fat food products [19]. For other *Gracilaria* spp., it was reported [8] that *G. edulis* showed an OHC of 1.64 g/g, which is very similar to the OHC values of the present study. Moreover, for *G. changii* [19] OHCs of 3.11 g/g at 24 °C and 1.17 g/g at 37 °C were reported. The low oil absorption capacity of red seaweeds is generally related to the hydrophilic nature of the changed polysaccharides (agar, carrageenan, fucans and alginates) of soluble dietary fiber [39]. The results of the present study for physicochemical properties confirmed that *G. corticata* and *G. edulis* could be considered as a source of food ingredients including proteins, dietary and soluble fiber [39].

## 4. Materials and Methods 

### 4.1. Sample Collection

Samples of the commercially important and commonly edible red seaweeds *G. edulis* and *G. corticata* were collected by hand from the Thondi Coast (Latitude: 9° 44′ N and Longitude: 79° 00′ E), Palk Bay, on the southeast coast of India. Freshly collected seaweeds were washed thoroughly in seawater and transported to the laboratory immediately. Epiphytes, sediment particles and other debris were removed by washing thoroughly using potable water and thoroughly washed with distilled water immediately after washing with potable water. Seaweeds were identified using a standard manual [40]. The voucher specimen was deposited in museum at Department of Oceanography and Coastal Area Studies, School of Marine Sciences, Alagappa University. Seaweeds were shade dried for 5 days at constant temperature of 25 °C ± 2 °C. Dried seaweed samples were powdered using a mechanical blender and stored at room temperature in an airtight container (Tarsons, Kolkatta, India) for further analysis within a maximum period of one week. Further the remaining powder sample stored at frozen condition (−20 °C) for future use.

### 4.2. Proximate Composition 

The proximate composition of each *Gracilaria* species was determined in all cases following Association of Official Analytical Chemists (AOAC) methods [41]. AOAC methods were employed to determine in *G. corticata* and *G. edulis* the ash content by heating at 550 °C for 24 h in a muffle furnace (AOAC, 930.05), moisture content by heating at 105 °C for 24 h (AOAC, 934.01), total fat by Soxhlet extraction with petroleum ether (AOAC 991.36) and protein by the Kjeldahl method (N × 6.25) (AOAC 981.10) [41]. The total carbohydrate content of *G. corticata* and *G. edulis* was determined by the phenol–sulfuric acid method [38]. All measurements were performed in triplicate for each seaweed and expressed as g/100 g seaweeds in a dry weight matter.

### 4.3. Isolation of Polysaccharides 

Polysaccharides were separated from *G. corticata* and *G. edulis* using 2 g samples of each seaweed according as previously described [42]. Powdered *G. corticata* and *G. edulis* were dissolved and homogenized with distilled water under constant stirring for 2 h at 100 °C. The residues obtained were then removed by centrifuging the sample at 6300 *g* for 10 min using a CPR 30 plus centrifuge (Remi Lab World, Mumbai, India). The obtained supernatant was then precipitated by addition of an ethanol-in-water solution (1:3 v/v), followed by subsequent washing with 30 mL of acetone. The precipitated polysaccharides were then collected and subsequently air-dried, re-dissolved in distilled water and washed with acetone. Afterwards, the collected polysaccharides were stored at −20 °C in a deep freezer (Blue Star, Mumbai, India) for further use. Isolation were performed in triplicate for each seaweed and results are expressed as g/100 g seaweeds in a dry weight basis.

### 4.4. Extraction of Crude Lipid and Determination of Fatty Acids Content

Portions of 500 mg each of powdered *G. corticata* and *G. edulis* were mixed with 5 mL of a chloroform:methanol solution (2:1 v/v), tightly covered with aluminum foil and kept at room temperature for 24 h. After this period, solutions were filtered through 11 µm Whatman No. 1 filter paper, and the filtered extract was placed in a pre-weighed and oven-dried beaker. The beaker was weighed with lipids, and the difference in weight was taken as total lipid content and expressed as a percentage [43]. Afterwards, an aliquot of the total lipids of each sample was used to determine the fatty acids content, based on a method published [6]. For this purpose, 0.45 g was introduced into a 10 mL volumetric flask, dissolved in hexane containing 50 mg of butylated hydroxytoluene per L and diluted to 10 mL with the same solvent. Afterwards, 2 mL of the solution was transferred into a quartz tube and evaporated by means of a nitrogen flow. Further, 1.5 mL of a 20 g/L solution of sodium hydroxide in methanol, covered with nitrogen, was added, capped tightly with a polytetrafluoroethylene-lined cap, mixed and heated in a water bath for 7 min. After the water bath, samples were cooled at room temperature, and 2 mL of boron trichloride-methanol solution was added; then, they were blanketed with nitrogen, capped tightly, mixed and heated in a water bath for 30 min. After this period, samples were cooled to 40–50 °C, and 1 mL of trimethylpentane was added; then, they were capped and shaken vigorously for at least 30 s. Immediately, 5 mL of saturated sodium chloride solution was added, then the samples were covered with nitrogen, capped and vortexed or shaken thoroughly for at least 15 s. The upper layer was allowed to become clear and then transferred to a separate tube. In the separate tube, the methanol layer was shaken once more with 1 mL of trimethylpentane and combined with trimethylpentane extracts. The organic solvent was then removed, and FAME were subjected to gas chromatography (GC), performed on a Perkin Elmer Clarus 580 gas chromatograph (Perkin Elmer, Gaithersburg, MD, USA) equipped with a flame ionization detector and an HP-5 capillary column (30 m × 0.25 mm). Initial temperature was maintained at 70 °C, then increased to 250 °C (10 °C/min); the injection temperature employed was 225 °C. Helium was used as carrier gas, with a flow rate of 1 μL/min. FAME peaks were identified by comparison of their retention times and quantified by comparison with individual calibration curves performed with a standard FAME mix (Supelco, Sigma-Aldrich, St Louis, MO, USA). Tricosanoic acid (C23; Sigma-Aldrich) was used as internal standard. Fatty acid composition was determined in triplicate for each seaweed and was expressed as g FAME/100 g total fat.

### 4.5. Protein Composition

Protein composition of *G. corticata* and *G. edulis* was determined based on a reversed-phase high performance liquid chromatography (HPLC) analysis method [6]. Two grams each of powdered seaweeds was mixed with phosphate buffer (pH 7.0) and centrifuged at 1200 *g* (Remi Lab World) for 20 min at 4 °C. The supernatant was collected, and protein content was precipitated by adding 10% v/v trichloroacetic acid (TCA). The protein pellet was resuspended in 1 N NaOH and hydrolyzed by heating the solution with 6 N HCl in a boiling water bath for 24 h. After incubation, the supernatant was collected by centrifuging the sample at 3500 *g* for 15 min. The supernatant was then filtered and neutralized by the addition of 1 N NaOH. The filtered supernatant was diluted to 1:100 (v/v) with deionized water. The sample was subjected to reversed-phase HPLC analysis (Lachrome Hitachi, Tokyo, Japan) with UV and fluorescence detectors. One µl of sample was injected into a Denali C18 5-mm column (4.6 mm × 150 mm) at 23 °C with detection at 254 nm. The mobile phase used was 20 mM sodium acetate/triethylamine (0.018% v/v) in phase A. The pH was adjusted to 7.2 using 1–2% acetic acid. In phase B, 20% of 100 mM sodium acetate (pH 7.2) with 1–2% acetic acid was used; 40% acetonitrile was used as the mobile phase. The protein composition was determined in triplicate for each seaweed and expressed as mg of amino acid/g of seaweed in a DW basis.

### 4.6. Determination of Mineral Content

Zn, Cu, Mg and Fe analysis of *G. corticata* and *G. edulis* was performed according to the European Standards, with minor modifications according to the method previously described [17]. One gram of homogenized seaweed sample was added to mixed reagent at a ratio of 5:2:1 (nitric acid:perchloric acid:sulfuric acid). Mineralization was performed on a hot plate at 50 °C for 30 min. After the end of digestion, 10 mL of 2 N HCL was added; digested solvents were filtered and made up to 25 mL with distilled water and stored at room temperature for further analysis [44]. Zn, Cu, Mg and Fe were determined by atomic absorption spectroscopy (AAS; Anton Paar-AAS, Graz, Austria). The metal standards were prepared and run to check the precision of the instrument throughout the analysis. Quality assurance and quality control protocols set by the US Environmental Protection Agency [45] for metal analysis were used. Quality assurance testing relied on the control of blanks and yield for the chemical procedure. Mineral content was determined in triplicate for each seaweed and expressed as mg/kg of DW seaweed.

### 4.7. Determination of Vitamin Content

The vitamin content of *G. corticata* and *G. edulis* was determined by the method previously described [20]; vitamin A or retinol (328 nm), vitamin B1 or thiamine monohydrate (420 nm), vitamin B2 or riboflavin (254 nm), vitamin B6 or pyridoxine HCl (254 nm), vitamin C or ascorbic acid, vitamin E or tocopheryl acetate (520 nm) and folic acid (550 nm) were determined by HPLC methods, and the results were compared with the respective standards retinyl acetate, thiamine monohydrate, riboflavin, pyridoxine HCl, ascorbic acid, tocopheryl, and folic acid (Sigma Aldrich). Vitamin content was determined in triplicate for each seaweed and was expressed as mg/g of DW seaweed.

### 4.8. Determination of Chlorophyll a, b and Carotenoids

Thin-layer chromatography (TLC) was used to screen chlorophyll *a* and *b* and carotenoid content in the *G. corticata* and *G. edulis* extracts [8]. The mobile phase contained methanol and chloroform (1:9). The sample (approximately 1 mg/mL) was spotted onto the TLC plates and air-dried. The spots were identified under long-wave and short-wave UV light, and also in an iodine chamber. The Rf value, which is the distance moved by the solute relative to the distance moved by the solvent, was calculated to find chlorophyll *a* and *b* and carotenoid content.

Afterwards, for chlorophyll *a* and *b* determination, 1 g each of *G. corticata* and *G. edulis* powder was extracted with 96% CH_3_OH, and the supernatant was collected by centrifugation at 1000 *g* for 1 min. After filtration, the supernatant was again filtered with Whatman No. 1 filter paper, centrifuged at 2300 *g* (Remi Lab World) for 10 min, and the absorbance of the collected supernatant was measured using a UV–Vis spectrophotometer [8]. The chlorophyll *a* and *b* contents were calculated using the following formulas in triplicate for each seaweed and expressed as µg/g of DW:Chlorophyll *a* = 15.65 (A_666_) − 7.340 (A_653_)(1)
Chlorophyll *b* = 27.05 (A_653_) − 11.21 (A_666_)(2)
where A_666_ = absorbance at 666 nm; A_653_ = absorbance at 653 nm. Carotenoid content was determined according to [46]. One gram each of *G. corticata* and *G. edulis* powder was extracted with 5 mL of acetone and incubated in the dark for 45 min, and the supernatant was collected by centrifugation at 10,000 *g* for 5 min. The supernatant was then stored in a refrigerator, and the extraction repeated with acetone until it became colourless. The supernatant was pooled and made up to 10 mL with acetone, and the absorbance of the collected supernatant was measured at 450 nm using a UV–Vis spectrophotometer (Shimadzu, Kyoto, Japan). Carotenoid content was calculated using the following formula in triplicate for each seaweed and expressed as µg/g of DW:Carotenoid content = A450/2500(3)
where 2500 is the extinction coefficient.

### 4.9. Physicochemical Properties

In order to determine the physicochemical properties of the *G. corticata* and *G. edulis* powder, the SWC, WHC and OHC were determined for each powder. The SWC of *G. corticata* and *G. edulis* was assessed based on a method previously described [38]. Briefly, 500 mg of *G. corticata* and *G. edulis* were taken and mixed with 20 mL of distilled water and stirred vigorously. The influence of temperature on SWC was determined by maintaining the tubes at 25 °C and 37 °C overnight. The SWC of *G. corticata* and *G. edulis* was calculated using the following formula and expressed as ml of swollen sample per g of DW:SWC = Initial volume of water (mL) − Volume of water after incubation (mL)(4)

The WHC of *G. corticata* and *G. edulis* was assessed by a modified method [38]. Briefly, 500 mg each of *G. corticata* and *G. edulis* was put into two sets of centrifuge tubes, and 20 mL of deionized water was added. The tubes were kept separately in an incubator shaker for 24 h at 25 and 37 °C. The supernatant was discarded after centrifuging the tubes at 12,000 *g* for 30 min (Remi Lab World). The wet weight of *G. corticata* and *G. edulis* was noted. The samples were dehydrated by keeping them in an oven at 160 °C for 2 h, and the dry weight of the sample was noted. The WHC was determined in triplicate for each seaweed and calculated using the following formula and expressed as the weight in g of water held by 1 g of DW sample:WHC = Wet weight of the sample (g) – Dry weight of the sample (g)(5)

The OHC of *G. corticata* and *G. edulis* was assessed according to previous methods [20,38]. About 3 g of dried *G. corticata* and *G. edulis* was taken in a tube and mixed with 10.5 g of corn oil. The tubes were placed in a shaker at room temperature for 30 min. The oil supernatant was collected by centrifugation at 3000 *g* for 30 min (Remi Lab World). The OHC was determined in triplicate for each seaweed and was determined using the following formula and expressed as the number of g of oil held by 1 g of DW sample:OHC = Initial volume of oil (g) − Volume of oil after incubation (g)(6)

### 4.10. Statistical Analysis

All results were expressed as mean ± SD. Paired sample *t*-test was used to compare composition values between *G. edulis* and *G. corticata*. One-way analysis of variance (ANOVA) and Duncan’s test were used to compare the effects of temperature on the physicochemical properties. All determinations were performed using SPPS version 14 (SPSS Science, Chicago, IL, USA). A positive significant variation was defined at the significance level of *p* < 0.05.

## 5. Conclusions

The polysaccharide content of the investigated dried red seaweeds (*G. corticata* and *G. edulis*) were found to be rich sources of polysaccharides. Because of the potential as prebiotics, antioxidant and anti-inflammatory compounds, carbohydrates from seaweeds has are compounds in high demand by consumers today. The physicochemical properties and proximate composition revealed that both *G. corticata* and *G. edulis* have appreciable levels of ash, protein, carbohydrate, fatty acid, essential and non-essential amino acid and vitamin content. It is suggested that both the seaweeds tested have great potential as potential food supplements and may be used in the food industry as a source of ingredients with an appreciable amount of nutritional value. Since both red seaweeds were found to be a good source of essential nutrients, their commercial value can be enhanced by marketing them as value-added products. However, depending on their composition, *G. edulis* and *G. corticata* have important differences that make it more adequate for certain cases. *G. edulis* showed higher concentration of essential amino acids, chlorophyll, vitamin B_2_ and Zn. Thus, it could be a good nutrient for low-protein diets or people whose need to reduce their oxidative status, because of its content in chlorophyll and Zn. Contrariwise, *G. corticata* showed higher PUFA content, carotenoids and minerals as Fe and Mg. Thus *G. corticata* is more adequate than *G. edulis* for people who need to reinforce their intake of such nutrients.

## Figures and Tables

**Table 1 molecules-24-02225-t001:** Proximate composition (g/100 g dry weight seaweed) *G. corticata* and *G. edulis*.

Parameter	*G. corticata*	*G. edulis*
**Moisture**	8.40 ± 0.65 ^b^	10.40 ± 0.69 ^a^
**Protein**	22.84 ± 0.87 ^b^	25.29 ± 0.67 ^a^
**Fat**	7.07 ± 0.33 ^a^	4.76 ± 0.73 ^b^
**Carbohydrates**	8.30 ± 1.89 ^a^	4.71 ± 0.60 ^b^
**Ash**	8.10 ± 0.49	7.36 ± 0.39
**Polysaccharides**	49.64 ± 3.89 ^a^	38.02 ± 4.32 ^b^

Values are mean ± standard deviation, *n =* 3. ^a–b^ values with different superscripts within the same line were significantly different.

**Table 2 molecules-24-02225-t002:** Fatty acids profile (g fatty acid methyl esters/100 g total fat) of *G. corticata* and *G. edulis*.

Parameter	*G. corticata*	*G. edulis*
**Palmitic acid**	1.22 ± 0.04 ^a^	0.63 ± 0.09 ^b^
**Margaric acid**	0.16 ± 0.25	0.15 ± 0.10
**Stearic acid**	1.31 ± 0.03 ^a^	0.93 ± 0.05 ^b^
**Oleic acid**	0.18 ± 0.12 ^b^	1.05 ± 0.05 ^a^
**Linoleic acid**	0.63 ± 0.04	0.65 ± 0.18
**α-Linolenic acid**	1.26 ± 0.04 ^a^	0.14 ± 0.04 ^b^
**Stearidonic acid**	0.21 ± 0.01	0.22 ± 0.05
**Docosohexaenoic acid**	0.48 ± 0.14 ^a^	0.12 ± 0.18 ^b^
**∑FA**	5.49 ± 0.30 ^a^	3.92 ± 0.13 ^b^

∑FA = Total fatty acids. Values are mean ± standard deviation, *n =* 3. ^a–b^ values with different superscripts within the same line were significantly different.

**Table 3 molecules-24-02225-t003:** Protein composition of *G. corticata* and *G. edulis*.

Amino Acid	*G. corticata*	*G. edulis*
**Aspartic acid**	14.37 ± 0.78	12.67 ± 0.64
**Glutamic acid**	2.54 ± 0.06	2.77 ± 0.15
**Asparagine**	1.45 ± 0.05 ^b^	1.89 ± 0.15 ^a^
**Serine**	2.23 ± 0.18 ^b^	2.73 ± 0.13 ^a^
**Glutamine**	2.01 ± 0.7 ^b^	2.42 ± 0.29 ^a^
**Glycine**	4.71 ± 0.18 ^a^	3.42 ± 0.27 ^b^
**Theronine ***	1.32 ± 0.09 ^b^	20.57 ± 0.62 ^a^
**Arginine ***	3.41 ± 0.30	3.33 ± 0.17
**Alanine**	21.11 ± 0.54 ^a^	1.46 ± 0.18 ^b^
**Cysteine**	1.49 ± 0.30	1.27 ± 0.06
**Tyrosine ***	1.25 ± 0.15 ^b^	2.50 ± 0.24 ^b^
**Histidine**	2.46 ± 0.27 ^a^	0.18 ± 0.02 ^a^
**Valine ***	0.16 ± 0.01	0.15 ± 0.02
**Methionine ***	8.73 ± 0.31 ^a^	4.98 ± 0.48 ^b^
**Isoleucine ***	2.53 ± 0.16 ^a^	1.22 ± 0.07 ^b^
**Phenylalanine ***	1.42 ± 0.17 ^b^	2.20 ± 0.10 ^a^
**Leucine ***	1.58 ± 0.35 ^a^	0.38 ± 0.02 ^b^
**Lysine ***	2.37 ± 0.27 ^a^	0.22 ± 0.03 ^b^
**Proline**	0.47 ± 0.20	0.46 ± 0.18
**Tryptophan**	1.00 ± 0.07 ^a^	0.59 ± 0.16 ^b^
**Total amino acids**	76.60 ± 5.14 ^a^	65.42 ± 3.58 ^b^
**Total EAAs**	22.76 ± 1.81 ^b^	35.55 ± 1.75 ^a^
**Total NEAAs**	36.14 ± 3.33 ^a^	29.86 ± 1.83 ^b^
**EAAs/Total AAs**	0.29 ± 0.35 ^b^	0.54 ± 0.48 ^a^
**EAAs/NEAAs**	0.62 ± 0.54 ^b^	1.19 ± 0.95 ^a^

Values are mean ± standard deviation, *n* = 3 expressed as mg/g seaweed in a dry weight basis. * EAAs: Essential amino acids; NEAAs: Non-essential amino acids. ^a–b^ values with different superscripts within the same line are significantly different.

**Table 4 molecules-24-02225-t004:** Mineral content (mg/kg) in *G. corticata* and *G. edulis*.

Minerals	*G. corticata*	*G. edulis*
**Zn**	31.52 ± 0.69 ^b^	42.73 ± 2.12 ^a^
**Cu**	11.42 ± 0.72 ^b^	14.61 ± 0.46 ^a^
**Mg**	463.23 ± 8.87 ^a^	89.56 ± 0.77 ^b^
**Fe**	1072.48 ± 20.97 ^a^	557.36 ± 0.57 ^b^

Values are mean ± standard deviation, *n* = 3 on dry weight basis. Macro minerals: Zinc (Zn), copper (Cu), magnesium (Mg) and iron (Fe). ^a–b^ values with different superscripts within the same line are significantly different.

**Table 5 molecules-24-02225-t005:** Vitamin composition (mg/g), chlorophyll and carotenoid content (µg/g) of *G. corticata* and *G. edulis*.

Name of Vitamins/Pigments	*G. corticata*	*G. edulis*
**Vitamin B_1_**	0.38 ± 0.02	0.36 ± 0.02
**Vitamin B_2_**	0.05 ± 0.01 ^b^	1.54 ± 0.07 ^a^
**Vitamin B_3_**	1.54 ± 0.39	1.10 ± 0.29
**Vitamin B_6_**	3.79 ± 0.30 ^b^	4.77 ± 0.23 ^a^
**Vitamin B_9_**	1.00 ± 0.07 ^a^	0.45 ± 0.06 ^b^
**Vitamin C**	14.66 ± 0.23	13.41 ± 0.57
**Vitamin A**	2.67 ± 0.30 ^a^	2.07 ± 0.06 ^b^
**Vitamin E**	1.40 ± 0.10	1.49 ± 0.10
**chlorophyll *a***	8.96 ± 0.39 ^b^	17.14 ± 0.55 ^a^
**chlorophyll *b***	7.74 ± 0.33	8.44 ± 0.63
**carotenoid**	12.82 ± 0.50 ^a^	2.99 ± 0.56 ^b^

Values are mean ± standard deviation, *n* = 3 on dry weight basis. ^a–b^ values with different superscripts within the same line are significantly different.

**Table 6 molecules-24-02225-t006:** Swelling capacity (SWC), water holding capacity (WHC) and oil holding capacity (OHC) of *G. corticata* and *G. edulis*.

Seaweeds	SWC (mL/g)		WHC (g/g)		OHC (g/g)	
	25 °C	37 °C	25 °C	37 °C	25 °C	37 °C
***G. corticata***	7.90 ± 0.32 ^bA^	5.70 ± 0.65 ^b^^B^	4.03 ± 0.39	3.96 ± 0.58	2.06 ± 0.24	1.84 ± 0.40
***G. edulis***	8.66 ± 0.53 ^aA^	7.70 ± 0.6 ^a^^B^	4.09 ± 0.32	3.64 ± 0.40	1.87 ± 0.28	1.91 ± 0.18

Values are mean ± standard deviation, *n* = 3 on dry weight basis. ^a–b^ values with different superscripts within the same column are significantly different between seaweeds. ^A–B^ values with different superscripts within the same line shows significant differences between temperatures.

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
