# Peer review of "Biochemical, Micronutrient and Physicochemical Properties of the Dried Red Seaweeds Gracilaria edulis and Gracilaria corticata"

_molecules, 2019, doi:10.3390/molecules24122225_

Round 1

Reviewer 1 Report

The Manuscript by Rosemary Thomas and co-authors (molecules-502963) entitled “Proximate, micronutrient and physicochemical  properties of red seaweeds Gracilaria edulis and  Gracilaria corticata” provides some information on nutritional compositions of two edible seaweeds of the genus Gracilaria.   

In my opinion, this manuscript can not be accepted for publication:

My major comments:

“2. Results”: it is very difficult to follow the data description and interpretation due to a combination of the results obtained and comparisons of the known findings (actually, “discussion”) which are in many cases not relevant (some examples are given below).

Why were these species used, and what was the reason to compare them?

The differences with the previous results of the papers, where the same parameters have been measured (e.g.,  in the case of cited paper by Sakthivel and Devi on G. edulis”), are explained by “However, the nutrient profile of seaweeds such as Gracilaria is influenced by different factors (line 62) such as seaweed species, habitat, maturity stage, season, water temperature and the sampling (line 63) conditions and method employed in the determinations [2,3]”, but no even approximate data on conditions of their growth are given. No data are given on the storage conditions of the dried material. Moreover, methodology should not be a reason for such (sometimes very large) differences.  

In general, when talking about the dried material, the comparison with lyophilised or fresh material is not relevant, at least, for lipids and fatty acids (although I understand that this is the way how the seaweeds are meant to be manufactured and put on the market but it should be somehow highlighted/specified in the title).

Methodology : 1)  “3.4. Extraction of crude lipid and determination of lipid profile” – the described procedure is not a method of Bligh, E.G.; Dyer, W.J. where partitioning between chl/meth and water is used; how was the solvent evaporated?

Are the data on total lipids or total FA present in Figure 1 (and why is it abbreviated as FFA – free fatty acids?); they should not be the same!

The total fatty acid content of G. corticata and G. edulis was 5.49±0.30 g/100g and 3.52±0.13 g/100g, 130 respectively – Text; and : Fatty acid composition was expressed as g FAME/100 g total fat. (line 310 in Methods).  This should be clear in Figure 1.

Method for FA derivatisation is not common and should not be described in such details: a relevant reference needed. What Internal standard was used?

Abstract: “G. corticata and G. edulis contain important fatty acids (line 21) including palmitic, stearic, oleic, linolenic, α-linolenic and moroctic acids”. Text: Red seaweeds are particularly rich in saturated C18 and C20 saturated fatty acids and PUFAs such as C20:5 and C20:4 which have nutritional applications that lead to their extensive use in food, feed, (line 125) cosmetic, biotechnological and pharmaceutical applications [24,25]. The lipid profile of G. corticata and G. edulis is shown in Figure 1. – (Lipids or FAs??? Means (?) and n-number is not given here) .Yes, it is well-known that Gracilaria seaweeds  are enriched with arachidonic acid (C20:4 n6) which determines their antibacterial properties. How can the disappearance of C20:4 and C20:5 be explained? If your determination of FA correct (what I doubt!), the seaweeds used by you do not have any nutritional value for FA consumption. In general,

Linolenic and alfa-linolenic acids are the same names for C18:3 n3. Probably you mistakenly called linoleic acid (C18:2n6) by linolenic acid or misidentified it. High levels of C17:0 and C18:4 are very unusual for these species, and 17:0 for plants in general. 

No DHA is found in Gracilaria (the cited paper by Ortoz et. al. is on Ulva (green) and Durvillaea (brown) algae, where it is found in very small amounts, so not relevant).

Fatty acid profiles are different from those given by Sakthivel and Devi for dried G. edulis (and cited by you).

Values for chlorophylls and carotenoid in mg/g in Table 3 and in ug/g in Text.

Methods for chlorophylls: absorbance used is 663 nm; different ones were used in equations.

Author Response

With respect to the comments from the Reviewer 1:

With respect to the comments about “it is very difficult to follow the data description and interpretation due to a combination of the results obtained and comparisons of the known findings (actually, “discussion”) which are in many cases not relevant”:

Many thanks for your comments. According to the suggestion from the Reviewer, in the revised version of the manuscript Results and Discussion section were placed separately. Some of the refences that Reviewers considered not adequate, were changed to other more adequate references.

With respect to the comments about why were chosen the two Gracilaria species:

Many thanks for your comments. The two Gracilaria species employed were selected in basis of their special abundance in Palk Bay. India. To clarify it, it was stated in the Introduction section the following paragraph:

Gracilaria edulis and G. corticata is abundantly available in almost all seasons in Palk Bay, Southeast coast of India rather than other Gracilaria sp. Both G. edulis and G. corticata are commercially important and commonly edible seaweeds in India. These two algae exhibited in a previous work high biological activities (proximate composition, antioxidant, antibacterial, and biopreservative effects in seafoods during preservation and extended shelf life than other Gracilaria species [Arulkumar et al., 2018]. Thus, the …”

Additionally, a reference was added to the references section.

With respect to the comments about “The differences with the previous results of the papers, where the same parameters have been measured (e.g.,  in the case of cited paper by Sakthivel and Devi on G. edulis”), are explained by “However, the nutrient profile of seaweeds such as Gracilaria is influenced by different factors (line 62) such as seaweed species, habitat, maturity stage, season, water temperature and the sampling (line 63) conditions and method employed in the determinations [2,3]”, but no even approximate data on conditions of their growth are given. No data are given on the storage conditions of the dried material. Moreover, methodology should not be a reason for such (sometimes very large) differences. 

Many thanks for your comments. Red seaweeds G. edulis and G. corticata commercially important and commonly edible algal samples were collected by hand picking from the Thondi Coast (Latitude: 9o 44’ N and Longitude: 79o 00E), Palk Bay, Southeast coast of India. Freshly collected seaweeds were washed thoroughly in seawater and transported to the laboratory immediately. Epiphytes, sediment particles and other debris were removed by washing thoroughly using potable water and thoroughly washed with distilled water immediately after washing with potable water. Seaweeds were identified using a standard manual [ha et al., 2009]. The voucher specimen was deposited in museum at Department of Oceanography and Coastal Area Studies, School of Marine Sciences, Alagappa University. Seaweeds were shade dried for 5 days at constant temperature of 25 oC ± 2 oC. Dried seaweed samples were powdered using mechanical blender and stored at room temperature in an airtight container (Tarsons, India) for further analysis within a maximum period of one week. Further the remaining powder sample stored at frozen condition (–20 oC) for future use.

With respect to the comments about “In general, when talking about the dried material, the comparison with lyophilised or fresh material is not relevant, at least, for lipids and fatty acids (although I understand that this is the way how the seaweeds are meant to be manufactured and put on the market but it should be somehow highlighted/specified in the title).”

Many thanks for your suggestions. We compared our result with frozen seaweeds. As you suggested we have highlighted in the title as “Biochemical, micronutrient and physicochemical properties of dried red seaweeds Gracilaria edulis and Gracilaria corticata”. The word “biochemical” was included following the instructions of another Reviewer.

With respect to the comments about Methodology: 1) “3.4. Extraction of crude lipid and determination of lipid profile” – the described procedure is not a method of Bligh, E.G.; Dyer, W.J. where partitioning between chl/meth and water is used; how was the solvent evaporated?

Many thanks for your comments. The solvent was evaporated by following procedure.

Portions of 500 mg each of powdered G. corticata and G. edulis were mixed with 5 ml of a chloroform:  methanol solution (2 : 1 v/v) and tightly covered with aluminum foil and kept at room temperature for 24 h. After this period, solutions were filtered through an 11-µm Whatman No. 1 filter paper, and the filtered extract were put in a pre-weighed and oven-dried beaker. The beaker was weighed with lipids, and the difference in weight was taken as total lipid content and expressed as a percentage (Folch, 1957).

With respect to the comments about are the data on total lipids or total FA present in Figure 1:

Many thanks for your comments. In fact, there is a mistake. All data reported are referred to total tatty acids, and not total lipids of FFA. It was corrected throughout the manuscript. Additionally, due to the comments from another Reviewer, Figure 1 was converted into a Table.

With respect to the need of specify in the Figure 1 that fatty acids are expressed as g FAME/100g and the seaweeds content of each specie:

Many thanks for your comments. According to the suggestion from the Reviewer, both data were included in the Table 1 legend.

With respect to the comments about the description of fatty acids derivatization:

Many thanks for your comments. Derivation process was performed based on the method published by Syad et al. (2013) and was cited in the text. Internal standard employed was tricosanoic acid (C23) and was also cited in the text.

With respect to the comments about Abstract: “G. corticata and G. edulis contain important fatty acids (line 21) including palmitic, stearic, oleic, linolenic, α-linolenic and moroctic acids”. Text: Red seaweeds are particularly rich in saturated C18 and C20 saturated fatty acids and PUFAs such as C20:5 and C20:4 which have nutritional applications that lead to their extensive use in food, feed, (line 125) cosmetic, biotechnological and pharmaceutical applications [24,25]. The lipid profile of G. corticata and G. edulis is shown in Figure 1. – (Lipids or FAs??? Means (?) and n-number is not given here) .Yes, it is well-known that Gracilaria seaweeds  are enriched with arachidonic acid (C20:4 n6) which determines their antibacterial properties. How can the disappearance of C20:4 and C20:5 be explained? If your determination of FA correct (what I doubt!), the seaweeds used by you do not have any nutritional value for FA consumption. In general, Linolenic and alfa-linolenic acids are the same names for C18:3 n3. Probably you mistakenly called linoleic acid (C18:2n6) by linolenic acid or misidentified it. High levels of C17:0 and C18:4 are very unusual for these species, and 17:0 for plants in general.  No DHA is found in Gracilaria (the cited paper by Ortoz et. al. is on Ulva (green) and Durvillaea (brown) algae, where it is found in very small amounts, so not relevant). Fatty acid profiles are different from those given by Sakthivel and Devi for dried G. edulis (and cited by you).

Thank you very much for you comment. In fact, there was important mistakes in the exposure of the results. In the revised version, Figure 1 was converted into a Table, and DHA values, that was not included in the Figure 1 (is the last value by mistake has not been included in the graph), was included in the table. Text in the abstract, results and discussion was accordingly modified, and more adequate references was included to compare the results of fatty acid profile with respect to those obtained in our work. Please note that Sakthivel and Pandima Devi (2013) did not report in his article on the DHA content of Gracilaria. Sakthivel and Devi collected G. edulis in the northern part of Palk Bay and analysed the fatty acid profiles. But in the present study, we collected G. edulis and G. corticata from Thondi coast which is located 43 km away from the sampling area of Sakthivel and Devi. Hence there is every possibility for the influence of environmental factors such as habitat, maturity stage, season, water temperature and nutrient levels of water on the fatty acids content between two localities. 

In any case, the content of DHA in our case was significantly lower than those were reported by other authors, and it is the reason for which DHA content were avoid in the original version. No EPA was detected in the present work in both G. corticata or G. edulis.

With respect to the comments about “Values for chlorophylls and carotenoid in mg/g in Table 3 and in µg/g in Text”.

Many thanks for your comments. The Reviewer are right and values for chlorophyll were changed to µg/g in Table 3.

With respect to the comments about “Methods for chlorophylls: absorbance used is 663 nm; different ones were used in equations”.

Many thanks for your comments. The Reviewer are right absorbance at 663 nm were deleted from the text. The values used in the equations were according to the method cited by Saktivel and Pandima Devi (2015) reference, that was cited in the revised version of the manuscript.

Reviewer 2 Report

Dear authors,

The proposed work is interesting but, from my point of view, it is proposed in incorrect manner.

Aim of the work is to evaluate both the chemical composition (including proximate composition,lipid profile, amino acids, vitamins, chlorophylls and carotenoids) and the physiochemicall properties of two edible red seaweeds.

There is a general lack of details and there are inaccuracies throughout the text (even in the abstract section). The statistical tets used in tables 1, 2, 3, 4, 5 appear to be student t test and not ANOVA tests (as reported in material and method section). Furthermore, there are inaccuracies in the tables and in the figure.

References are sometimes inappropriate (for example, 3, 4, 20, 21 and 42) and sometimes not used appropriately (ref 15).

Sampling procedures are poorly described in the materials and methods section.

Another comment regards the nutritional profile: in my opinion data should be commented considering the RDA values and the serving portions.

In details:

Line 22 I think that stearidonic acid shold be a better term than moroctic acid;

Line 26-27 Please ceck the sentence concerning oil-holding capacity;

Line 45 ref 4 cited is not appropriate;

Line 53 ref 2 cited is not appropriate;

Line 70 As a first paragraph of the work I would put proximate and lipid composition;

Line 80-81. I don’t understand the sentence; is the atmospheric pressure the most important factor for extraction?

Section 2.2 The table relating to the proximate and lipid c composition is missing. I think it should be added to help the reader. What do the authors think?

Line 86-90. It is difficult to compare the different content in water with other scientific works as different methods are used to dry the samples. What do the authors think?

Line 98-100 Rference 20 and 21 are not appropriate.

 Figure 1. Fatty acid (%)…of what? Please complete the sentence.

Line 138-139. Are these seaweeds rich in MUFAs and PUFAs? I am not sure

Line 147-164 authors should choose a unit of measurement for amino acids and explain what the contents expressed in table 1 refer to. Furthermore, which statistical test was used to compare the data presented in the tables? ANOVA test as reported in materials and methods section (tab1, 2, 3, 4, 5)?

Line 175-176. Please ceck the sentence for Zn.

Table 3. please cech the content of vitamin B2 and its SD for G. corticata

Line 212-213. Values reported in ref 15 are on fresh weight;

Table 4; Please ceck the footnotes….”the same raw”?

Section 3.1. sapling should be described in more details; Furthermore, the samples should be washed with distilled water and not fresh water (line 263) as the analysis on the mineral content could be affected by this procedure.

Line 362-363. Please The equations should be written more clearly.

Author Response

With respect to the comments from the Reviewer 2:

With respect to the comment about “there is a general lack of details and there are inaccuracies throughout the text (even in the abstract section). The statistical test used in tables 1, 2, 3, 4, 5 appear to be student t test and not ANOVA tests (as reported in material and method section). Furthermore, there are inaccuracies in the tables and in the figure.”

Many thanks for your comments. Statistical comparisons were made by t-test, with the exception of the influence of temperature in the physicochemical properties, with were performed by ANOVA. It was corrected in the revised version of the manuscreipt. Inaccuracies in units of Tables were modified accordingly the observations of the Reviewers. Figure 1 was deleted and converted into a Table in the revised version of the manuscript.

With respect to the comment about “References are sometimes inappropriate (for example, 3, 4, 20, 21 and 42) and sometimes not used appropriately (ref 15).”

Many thanks for your comments. According to the suggestion from the Reviewer, all references cited were changed to other more appropriate according to the paragraph that they are referring. All references cited (3,4,20, 21 and 42) were deleted and modified by other more adequate references. Reference number 15 in the old version (Saktivel and Pandima Devi (2015) were revised when cited in the revised version of the manuscript.

With respect for the comments about sampling procedures are poorly described in the materials and methods section.

Thank you for your comment. Accordingly, in the revised version of the manuscript more information about sampling was included.

With respect to the comments about the nutritional profile: in my opinion data should be commented considering the RDA values and the serving portions.

Thank you for your comment. The portion size is not knowledge by us. The nutritional information is usually stated as the content in 100 g of food. Dietary recommendations are different depending on the country or geographical area and because in most countries are not mandatory their inclusion in the nutritional labelling. Thus, because both rations and recommendations can vary, we think that the inclusion of this information could confuse the reader.

With respect to the comments about “Line 22 I think that stearidonic acid shold be a better term than moroctic acid”

Many thanks for your comments. According to the suggestion from the Reviewer, “morictoc acid” was changed to “stearidonic acid” in the revised version of the manuscript.

With respect to the comments about “Line 26-27 Please check the sentence concerning oil-holding capacity”

Thank you for your comment. In fact, there was a mistake than was corrected in the revised version of the manuscript. In the revised version, the phrase “The water- and oil-holding capacities of G. corticata were found to be higher than those of G. edulis, and oil-holding capacity was very similar in both seaweeds” was changed to “The swelling capacity of G. edulis were found to be higher than those of G. corticata, whereas both water- oil-holding capacity was similar in both seaweeds.

With respect to the comments about “Line 45 ref 4 cited is not appropriate”

Thank you for your comment. According to the suggestion of the Reviewer, reference of Gracilaria composition was changed to Debbarama et al. (2016) and a new reference was introduced to reinforce the affirmation about DHA and cardiovascular diseases:

Meyer, B.J., de Groot, R.H.M. (2017). Effects of omega-3 long chain polyunsaturated fatty acid supplementation on cardiovascular mortality: The importance of the dose of DHA. Nutrients, 9, 1305.

With respect to the comments about “ref 2 cited is not appropriate”

Thank you for your comment. According to the suggestion of the Reviewer, reference of Gracilaria aminoacids composition was changed to composition was changed to Sakthivel and Pandima Devi (2015).

With respect to the comments about “as a first paragraph of the work I would put proximate and lipid composition”

According to the suggestion of the Reviewer, proximate and fatty acids composition was placed in materials and method section prior to polysaccharides content. The same way was follows in results and discussion sections.

With respect to the comments about line 80-81. I don’t understand the sentence; is the atmospheric pressure the most important factor for extraction?

Thank you for your comment. In fact, this phrase is confounding and in the revised version of the manuscript it was changed to “Additionally, the variations in the polysaccharide content of Gracilaria can vary depending on atmospheric temperature at the time of extraction”

With respect to the comments about the lack of a Table related to proximate and fatty acids content:

Thank you for your comment. According to the suggestion of the Reviewer, in the revised version of the manuscript, it was included a new table (Table 1) related to proximate and fatty acid composition.

With respect to the comments about: It is difficult to compare the different content in water with other scientific works as different methods are used to dry the samples. What do the authors think?

Thank you for your comment. In India, almost all the commercially important seaweeds are sold in dried condition for local companies and export. Hence, we aimed to analyses the commercially available form of seaweeds for this study. Evidently, when methods to dry samples are different, the results of moisture content (as well as other parameters) can vary, but in global terms these variations affect their nutritional properties in a discrete way.

With respect to the comments about References 20 and 21 are not appropriate:

Thank you for your comment. According to the suggestion of the Reviewer, in the revised version of the manuscript references 20 and 21 were deleted and a new reference about the human health benefits of dietary fibre was included:

Kendall, C.W.C.; Esfahani, A., Jenkins, D.J.A. The link between dietary fibre and human health. Food Hydrocoll 2010, 24, 42-48.

With respect to the comments “Figure 1. Fatty acid (%)…of what? Please complete the sentence”

Thank you for your comment. According to the suggestion of the Reviewer, is was specified in the new Table that fatty acids are expressed as g fatty acid methyl esters/100 total fat.

With respect to the comments about “are these seaweeds rich in MUFAs and PUFAs? I am not sure”:

Thank you for your comment. In the original version of the manuscript, in Figure 1 a mistake occurred when we graph the results and DHA content (the last data) were omitted. In the revised version, it was included in Table 1. In view of the results obtained, in fact, these seaweeds are not rich in MUFA (only contains oleic acid), only in PUFA. It was also corrected in the revised version of the manuscript.

With respect to the comments about “Line 147-164 authors should choose a unit of measurement for amino acids and explain what the contents expressed in table 1 refer to. Furthermore, which statistical test was used to compare the data presented in the tables? A Thank you for your comment. ANOVA test as reported in materials and methods section (tab1, 2, 3, 4, 5)?”

Thank you for your comment. In the revised version of the manuscript is was specified that amino acid results are expressed in mg/g seaweed on a dry weight basis. The statistical comparison was by t-test for comparison between seaweeds and ANOVA for the determination of temperature in physicochemical properties. It was corrected in the revised version of the manuscript.

With respect to the comments about Please check the sentence for Zn.

Thank you for your comment. In fact, the lower content found in this work were for Mg, and not for Zn, with respect to other works. In the revised version of the manuscript, it was clarified accordingly.

With respect to the comments about please check the content of vitamin B2 and its SD for G. corticata

Thank you for your comment. In fact, it was a mistake. In the revised version of the manuscript, the vitamin B2 content was changed from “0.005” mg/g to “0.05”.

With respect to the comments about Values reported in ref 15 are on fresh weight:

Thank you for your comment. In fact, most parameters cited in reference 15 was cited on a dray weight basis but chlorophylls were expressed on a fresh weight basis. Thus, the results of the two works are Hardly comparable, and consequently, this comparison was deleted in the revised version of the manuscript.

With respect to the comments about “Table 4; Please check the footnotes….”the same raw”?”

Thank you for your comment.  Is not “raw”, but “row”. Perhaps the term is confusing. In the revised version of the manuscript, we change the word “row” to “line”.

With respect to the comments about Section 3.1. sapling should be described in more details; Furthermore, the samples should be washed with distilled water and not fresh water (line 263) as the analysis on the mineral content could be affected by this procedure.

Many thanks for your comments. Based on your suggestions, sampling details were given:

Red seaweeds G. edulis and G. corticata commercially important and commonly edible algal samples were collected by hand picking from the Thondi Coast (Latitude: 9o 44’ N and Longitude: 79o 00E), Palk Bay, Southeast coast of India. Freshly collected seaweeds were washed thoroughly in seawater and transported to the laboratory immediately. Epiphytes, sediment particles and other debris were removed by washing thoroughly using potable water and thoroughly washed with distilled water immediately after washing with potable water. Seaweeds were identified using the standard manual of Umamaheshwara Rao, (1987); Jha et al. (2009). The voucher specimen was deposited in museum at Department of Oceanography and Coastal Area Studies, School of Marine Sciences, Alagappa University. Seaweeds were shade dried for 5 days at constant temperature of 25 oC ± 2 oC. Dried seaweed samples were powdered using mechanical blender and stored at room temperature in an airtight container (Tarsons, India) for further analysis within a maximum period of one week.

With respect to the comments about Line 362-363. Please The equations should be written more clearly.

Many thanks for your comments. According to the suggestion from the Reviewer were included a description of the meaning of A666= absorbance at 666 nm and A653= absorbance at 653 nm.

Reviewer 3 Report

Dear Authors,

Manuscript entitled “Proximate, micronutrient and physicochemical properties of red seaweeds Gracilaria edulis and Gracilaria corticata”, is an original research study but in current version does not adhere to the standards. Discussion of the results is limited and in my opinion the results from red seaweeds should be compared also with other alga or functional powders.  Generally, the study should be reconsidered for publication after minor revision.

Major:

1.                  What is the main aim of the study? Comparing of 2 seaweeds or functional powders. If you state that chemical composition is strongly diversified by conditions (L62-67- temperature, salinity etc.) maybe just compare 2 functional powders (especially in the light of physical properties).

2.                  What is the novelty of the study? In the current version seems that only the place where seaweeds were gathered.

3.                  Abstract is usually stand-alone. Please rewrite it to make its self-explanatory. Without details from the main body of the manuscript, it is incomprehensible. Ls26-27 do not makes sense.

4.                  Please add Table with proximate analysis.

5.                  Add statistical analysis for results

6.                  Prepare references according to Guide.

7.                  Discussion of the results is very limited and should be significantly improved.

Minor:

L40 Unclear. Free amino acids? Maybe: proteins containing ….

L48 What is accumulated? Proteins, free amino acids … ???

L69 and Discussion

L70 What do you mean by polysaccharides?

L72 DW explain

L83-84; L139-145; L218, L209. Move to Introduction

L94 cursive

L111. Unify nomenclature. There are some inaccuracies. E.g. polysaccharides (45%) vs.  Total carbohydrates (8%)??????

L146 It is protein composition. Please change it (also in the rest of the manuscript).

L148 mg/kg?

Please explain differences between protein content and amino acids. See e.g. L103 vs. L148.

G.c 22.8- 76  vs. G.e 25 – 65 low-high????

Table 1. S.No. what is it? mg/g of what?

Fig 1. Fatty acids in % of what?

L168.   Do you have any information about heavy metals in your powders.

Table 3. mg/g or ug/g?? see L361

L261. Please provide more information about powders obtaining. Drying how long, Time of seaweeds gathering. Many of components (e.g. vitamins) are sensitive and their content change significantly after draying.

L405. Did you study dietary fibre?

Conclusion should be general but in fact, you compare 2 seaweeds. Please include some observation in this part.  

Author Response

With respect to the comments from the Reviewer 3:

With respect to the comments about Discussion of the results is limited and in my opinion the results from red seaweeds should be compared also with other alga or functional powders.  Generally, the study should be reconsidered for publication after minor revision.

Many thanks for your comments. In the revised version of the manuscript, we divided the section Results and Discussion into separate subheadings and added various new references to compare and discuss with our results.

With respect to the comments about what is the main aim of the study? Comparing of 2 seaweeds or functional powders. If you state that chemical composition is strongly diversified by conditions (L62-67- temperature, salinity etc.) maybe just compare 2 functional powders (especially in the light of physical properties).

Many thanks for your comments. In the revised form of the manuscript we include a new paragraph about the aim of the study. In concrete:

Gracilaria edulis and G. corticata is abundantly available in almost all seasons in Palk Bay, Southeast coast of India rather than other Gracilaria sp. Both G. edulis and G. corticata are commercially important and commonly edible seaweeds in India. These two algae exhibited in a previous work high biological activities (proximate composition, antioxidant, antibacterial, and biopreservative effects in seafoods during preservation and extended shelf life than other Gracilaria species (Arulkumar et al., 2018). Thus, the…”

With respect to the comments about what is the novelty of the study? In the current version seems that only the place where seaweeds were gathered.

Many thanks for your comments. In fact, the aim and method employed is like in example the article published by Sakthivel and Pandima Devi (2015) regarding G. edulis. However, please note that various of the results found are very different than those obtained in the cited work and consequently, it is important to the reader to highlight that depending on the Gracilaria species, origin, season, etc the nutritional profile and physical properties are different.

With respect to the comments about abstract is usually stand-alone. Please rewrite it to make its self-explanatory. Without details from the main body of the manuscript, it is incomprehensible. Ls26-27 do not makes sense.

Many thanks for your comments. Please note that the extension of the abstract is restricted by Molecules guidelines. In any case, we changed the content of the abstract to make it self-explanatory and especially corrected the content of lines 26-27.

With respect to the comments about please add table with proximate analysis.

Many thanks for your comments. According to the suggestions from the Reviewer a new table (Table 1) was added to the revised version of the manuscript. Accordingly, data originally included in Figure 1 was also transferred to the same Table.

With respect to the comments about add statistical analysis for results.

Many thanks for your comments. In the materials and method sections, it was specified that only results with differences reached a significance level lower than 0.05 were considered. Taking into account the high number of data compared in the present work, we think that the inclusion of P values for all comparison will make the manuscript too long and hard to read. However, if you consider that individual P values are indispensable, we would be delighted to include you in the next round of review.

With respect to the comments about prepare references according to guide.

Many thanks for your comments. The references included in the references section are according to Molecules guidelines. We assume that the Reviewer refer to the references in the text. In the revised version of the manuscript, the names of authors were omitted and cited only by numbers in alphabetical order, according to Molecules guidelines.

With respect to the comments about Discussion of the results is very limited and should be significantly improved.

Many thanks for your comments. In the revised version of the manuscript, it was placed the discussion separately from the results and more references and discussion was added.

With respect to the comments about L40 Unclear. Free amino acids?

Many thanks for your comments. According to the suggestions from the Reviewer, the expression “free amino acids” was changed to “protein”.

With respect to the comments about What is accumulated? Proteins, free amino acids … ???

Many thanks for your comments. According to the suggestions from the Reviewer, in the revised version of the manuscript is was clarify that with the “compounds” was are referring to “aminoacids”.

With respect to the comments about L69 and Discussion

Many thanks for your comments. In the revised version of the manuscript results and discussion were spitted, so this comment is not applied.

With respect to the comments about what do you mean by polysaccharides?

Many thanks for your comments. With polysaccharides we are referring to are the polymers composed of at least 10 monosaccharides linked in union by glycosidic bonds. This definition was also included in the revised version of the article to improve the discussion section.

With respect to the comments about explain DW

Many thanks for your comments. In the revised version of the manuscript is was stated the meaning of DW (dry weight) the first time that it appears in the text. The same was made for the other abbreviations. In the original version it have been shown in the wrong order, as they were defined in the section on materials and methods, which according to the format of the journal, appears after the discussion.

With respect to the comments about L83-84; L139-145; L218, L209. Move to Introduction

Many thanks for your comments. According to the suggestions from the Reviewers, al the cited phrases and paragraphs were moved from the discussion to the introduction.

With respect to the comments about L94 cursive

Many thanks for your comments. According to the suggestions from the Reviewers, changii, as well as other scientific names of seaweeds, were written in italics.

With respect to the comments about Unify nomenclature. There are some inaccuracies. E.g. polysaccharides (45%) vs.  Total carbohydrates (8%)??????

Many thanks for your comments. Please note that carbohydrates and polysaccharides are different concepts, because polysaccharides can include in their structure other compounds that are not carbohydrates. Please note also that in most published articles about seaweeds composition, often polysaccharides are higher than carbohydrates. In example:

Sakthivel, R.; Devi, K.P. Evaluation of physicochemical properties, proximate and nutritional composition of Gracilaria edulis collected from Palk Bay. Food Chem 2015, 174, 68-74.

With respect to the comments about L146 It is protein composition. Please change it (also in the rest of the manuscript).

Many thanks for your comments. According to the suggestion from the Reviewer, “amino acid content” was changed to “protein composition” throughout the manuscript.

With respect to the comments about L148 mg/kg?

Many thanks for your comments. According to the observation from the Reviewer, in the revised version of the manuscript “mg/kg” was changed to “mg/g”.

With respect to the comments about Please explain differences between protein content and amino acids. See e.g. L103 vs. L148. G.c 22.8- 76  vs. G.e 25 – 65 low-high????

Many thanks for your comments. Please note that the units are different. Protein content is expressed as g/100 g, whereas total amino acids are expressed as mg/g. If they are converted to the same units, in example to G. corticata, protein content (22.8 g/100g) is equivalent to 228 mg/g, whereas total amino acids represent 76 mg/g. If fact, amino acid content is lower than those obtained for protein, but it should be considered that protein method employed is based in the quantification of N, that can be present in other compounds that are not amino acids.

With respect to the comments about Table 1. S.No. what is it? mg/g of what?

Many thanks for your comments. According to the observation from the Reviewer, the first column from Table 1 (now Table 2) was deleted. It was also specified in the footnote that mg/g is related to mg/g seaweed in a dry weight basis.

With respect to the comments about Fig 1. Fatty acids in % of what?

Many thanks for your comments. According to the comments from the Reviewers, Figure 1 was deleted and converted into a Table (Table 1). It was specified that units are expressed in g fatty acid methyl esters/100 total fat.

With respect to the comments about Do you have any information about heavy metals in your powders.

Many thanks for your comments. No, at this time we did not determined heavy metals content. It is a pity because they would undoubtedly have been interesting results that would improve the present manuscript

With respect to the comments about Table 3. mg/g or ug/g?? see L361

Many thanks for your comments. Both units are present in Table 3 (now Table 4). In the revised version of the manuscript are specified that vitamin composition is expressed in mg/g, whereas chlorophyll and carotenoid content is expressed in µg/g.

With respect to the comments about Please provide more information about powders obtaining. Drying how long, Time of seaweeds gathering. Many of components (e.g. vitamins) are sensitive and their content change significantly after draying.

Many thanks for your comments. According to the suggestion from the Reviewer, in the revised version of the manuscript is was included the following paragraph:

“Seaweeds were shade dried for 5 days at constant temperature of 25 oC ± 2 oC. Dried seaweed samples were powdered using mechanical blender and stored at room temperature in an airtight container (Tarsons, India) for further analysis within a maximum period of one week.”

With respect to the comments about L405. Did you study dietary fibre?

Many thanks for your comments. According to the suggestion from the Reviewer, the term “dietary fibre” was deleted from the conclusion section because was not determined in the present work.

With respect to the comments about Conclusion should be general but in fact, you compare 2 seaweeds. Please include some observation in this part. 

Many thanks for your comments. According to the suggestion from the Reviewer, it was included in the conclusion the following paragraph:

“However, depending on their composition, G. edulis and G. corticata have important differences that make it more adequate for certain cases. G. edulis showed higher concentration of essential amino acids, chlorophyll, vitamin B2 and Zn. Thus, it could be a good nutrient for low-protein diets or people whose need to reduce their oxidative status, because of its content in chlorophyll and Zn. Contrariwise, G. corticata showed higher PUFA content, carotenoids and minerals as Fe and Mg. Thus G. corticata is more adequate than G. edulis for people who need to reinforce their intake of such nutrients.”

Reviewer 4 Report

Comment 1-

The title of the manuscript must be changed to avoid copy titles, maybe change proximate by chemical or biochemical composition:

Proximate, micronutrient and physicochemical properties of red seaweeds Gracilaria edulis and  Gracilaria corticata

Evaluation of physicochemical properties, proximate and nutritional composition of Gracilaria edulis collected from Palk Bay (author Pandima Devi)

Comment 2

Revise the abstract, to avoid the exaggerated use of the word: source

(e.g First sentences)

Comment 3

First reference is related to Porphyra. Authors are advised to change that reference for the text

 very important natural resources from oceans that are employed as human foods 33 and animal feeds in their whole form, and as sources of polysaccharides (mainly alginates, 34 carrageenans and agar), carotenoids, lipids, vitamins, minerals, dietary fiber, proline and amino acids 35 for their use in food and pharmaceutical industry [1].

 Comment 4

Among red algae, the genus Gracilaria includes a broad diversity of valuable contents for human  nutrition. Its lipid content is low (1–5% dry matter, DM), but it contains docosahexaenoic acid (DHA)

22:5??

which is recognized as the most important n-3 polyunsaturated fatty acid (PUFA) to reduce the risk  of cardiovascular diseases [4].

4. Ortiz, J.; Romero, N.; Robert, P.; Araya, J.; Lopez-Hernández, J.; Bozzo, C.; Navarrete, E.; Osorio, A.; Rios, A. Dietary fiber, amino acid, fatty acid and tocopherol contents of the edible seaweeds Ulva lactuca and Durvillaea antarctica. Food Chem 2006, 99, 98-104.

This reference is green and brown algae. More than relevance of 22:5 fA is the 20:4 and 20:5, not refered by authors. The text must be changed to include these fatty acids from red algae, rather than DHA, not relevant for red algae. See references:

Evaluation of physicochemical properties, proximate and nutritional composition of Gracilaria edulis collected from Palk Bay

Ravi Sakthivel, KasiPandima Devi

Distribution of eicosapentaenoic and arachidonic acids in different species of Gracilaria

S.V.Khotimchenko, V.E.Vaskovsky, V.F.Przhemenetskayaa

Valorization of Lipids from Gracilaria sp. through Lipidomics and Decoding of Antiproliferative and Anti-Inflammatory Activity

Elisabete Da Costa 1, Tânia Melo 1, Ana S. P. Moreira 1, Carina Bernardo 2, Luisa Helguero 2, Isabel Ferreira 3, Maria Teresa Cruz 3, Andreia M. Rego 4, Pedro Domingues 1, Ricardo Calado 5, Maria H. Abreu 4 and Maria Rosário Domingues

The Red Seaweed Gracilaria gracilis as a Multi Products Source

Matteo Francavilla 1,2,*, Massimo Franchi 2, Massimo Monteleone 1 and Carmela Caroppo

Comparative evaluation and selection of a method for lipid and fatty acid extraction from macroalgae.

Kumari P1, Reddy CR, Jha B

 Biochemical composition and physicochemical properties of two red seaweeds (Gracilaria fisheri and G. tenuistipitata) from the Pattani Bay in Southern Thailand

Ommee Benjama* and Payap Masniyom

Comment 5

Regarding bioactivities of different compounds form Gracilaria,, authors should include some findings from:

The Red Seaweed Gracilaria gracilis as a Multi Products Source

Matteo Francavilla 1,2,*, Massimo Franchi 2, Massimo Monteleone 1 and Carmela Caroppo

Bioactivities from marine algae of the genus Gracilaria, Cyntia Layse de Almeida

Comment 6

Results                          

Results, contain discussion! Discussion must be mentioned as a topic.

Comment 7

g/100g, respectively, on a DW basis. Define first time DW

SWC, WHC and OHC define the first time used

G. edulis italicize the algae number and review it in the manuscript

Comment 8

The moisture content of G. corticata and G. edulis was 8.4±0.65 g/100g and 10.4±0.69 g/100g,.

Include spaces: 8.4 ± 0.65 g/100g

Comment 9

Line 85, Lipid profile:

Review lipid content and fatty acid contents

The content of lipid is high, as well as the content of fatty acids. Please consider revision of units and see literature, where lipid content is about less than 1%

The fatty acids 20:4 and 20:5 were not determined, rather than 18:4 (morotic FA), these 2 fatty acids assign Gracilaria sp.

See examples:“Reveal the presence of dietary fibre (8.9 ± 0.62% DW), carbohydrate (101.61 ± 1.8 mg/g DW), crude protein (6.68 ± 0.94 mg/g DW) and lipid content (8.3 ± 1.03 mg/g DW)...”

Evaluation of physicochemical properties, proximate and nutritional composition of Gracilaria edulis collected from Palk Bay

Ravi Sakthivel, KasiPandima Devi

Valorization of Lipids from Gracilaria sp. through Lipidomics and Decoding of Antiproliferative and Anti-Inflammatory Activity

Elisabete Da Costa 1, Tânia Melo 1, Ana S. P. Moreira 1, Carina Bernardo 2, Luisa Helguero 2, Isabel Ferreira 3, Maria Teresa Cruz 3, Andreia M. Rego 4, Pedro Domingues 1, Ricardo Calado 5, Maria H. Abreu 4 and Maria Rosário Domingues 1,

Comparative evaluation and selection of a method for lipid and fatty acid extraction from macroalgae.

Kumari P1, Reddy CR, Jha B

 Biochemical composition and physicochemical properties of two red seaweeds (Gracilaria fisheri and G. tenuistipitata) from the Pattani Bay in Southern Thailand

Ommee Benjama* and Payap Masniyom

Also include the table with absolute amount and percentage of fatty acids,  and change the legend of the fatty acid profile by the correspondant fatty acids chain : C16:0 instead of palmitic acid. If statistic was performed, results should be included . After correct the values as mentioned before, correct the graph.

Comment 10

In. 3 Materials and Methods

Provide information about replicates in all subsections.

Comment 11

Line 277- 3.3 section

Proximate composition, include brief description of the core methods used following AOAC methods, is easy to understand the analytical approach.

Regarding fatty acids and amino acids quantification, please include the internal standards and how absolute quantification was performed ( was based on calibration curves for the different fatty acids?)

Comment 12

Line 3.8- Pigments

Chlorophyll determination, is not clear how HPLC and UV contents were achieved: The equation/formulas were determined by authors ? Or are based in literature?

Chlorophyll a = 15.65 (A666) – 7.340(A653)  

Chlorophyll b = 27.05 (A653) – 11.21(A666)

how achieve the formula

Why not include other minerals than the 4 elements described? Only that 4 were analysed?

Comment 13

398. Statistics:

Variance (ANOVA) and Duncan’s 399 test were used to compare the effects of temperature on the physicochemical properties. Why do authors mention variance analysis if it was not mentioned in the comparison of results?Were is the information gathered from significance analysis

Comment 14

Line 402- Conclusions must be re-written to avoid similarities with previews reported works.

Author Response

With respect to the comments from the Reviewer 4:

With respect to the comments about the need to change the tittle:

According to the suggestion from the Reviewer, the word “proximate” was changed to “biochemical”.

With respect to the comments about the need to avoid the exaggerated use of “source” in the abstract:

According to the suggestion from the Reviewer, the word “source” was deleted from the abstract section.

With respect to the comments about the need to change the reference Cao et al. (2016).

The text cited is regarding all seaweed and not for Gracilaria, and consequently, the reference employed could be considered adequate. However, according to the comments from the Reviewer, in the revised version of the manuscript we change this reference to Debbarama et al. (2016).

With respect to the comments about the need to change references in “Among red algae, the genus Gracilaria includes a broad diversity of valuable contents for human nutrition. Its lipid content is low (1–5% dry matter, DM), but it contains docosahexaenoic acid (DHA) 22:5??which is recognized as the most important n-3 polyunsaturated fatty acid (PUFA) to reduce the risk of cardiovascular diseases [4].”

According to the suggestion of the Reviewer, reference of Gracilaria composition was changed to Debbarama et al. (2016). Additionally, two of the references recommended by the Reviewer were included:

Khotimchenko, S.V., Vaskovsky, V.E., Przhemenetskaya, V.F. (1991). Distribution of eicosapentaenoic and arachidonic acids in different species of Gracilaria. Phytochemistry, 30, 207-209.

Benjama, O., Masniyom, P. (2012). Biochemical composition and physicochemical properties of two red seaweeds (Gracilaria fisheri and G. tenuistipitata) from the Pattani bay in Sothern Thailand. Songklanakarin J Sci Technol 34, 223-230.

Additionally, a new reference was introduced to reinforce the affirmation about DHA and cardiovascular diseases:

Meyer, B.J., de Groot, R.H.M. (2017). Effects of omega-3 long chain polyunsaturated fatty acid supplementation on cardiovascular mortality: The importance of the dose of DHA. Nutrients, 9, 1305.

With respect to comments about to incorporate information about bioactivities of different compounds from Gracilaria:

Many thanks for your comments. According to the suggestion from the Reviewer, it was included in the references section the recommended article and information from the cited article was incorporated to the discussion of the current manuscript.

With respect to comments about results contains discussion and must be mentioned as a topic:

Many thanks for your comments. According to the suggestion from the Reviewer, in the revised version of the manuscript, results and discussion was stated separately.

With respect to comments about abbreviations and italicize the name of seaweed:

Many thanks for your comments. According to the suggestion from the Reviewer, abbreviation and seaweeds names were checked and corrected in the revised version of the manuscript.

With respect to comments about Include spaces: 8.4 ± 0.65 g/100g

Many thanks for your comments. According to the suggestion from the Reviewer, spaces were included in the results throughout the manuscript.

With respect to comments about lipid profile:

Many thanks for your comments. Please note that the lipid content was performed on dry basis seaweeds, and not for the fresh seaweeds. We performed the analysis on a dry weight basis because is the usual form in which seaweeds are commercialized. Please note that one of the articles cited by you (Sakthivel and Pandima Devi, 2015) found a lipid content of 8.3 in dried G. edulis (higher than those obtained in the present work). Other recent work (Chan and Matanjum, 2017) found in G. changii a lipid content of 3.3 %, and there are other works whose found variable lipid contents. The presence of DHA, accidentally omitted in Figure 1 in the original version of the manuscript, was included in Table 1 in the revised version. No EPA was found in the present study. EPA content is very variable in Gracilaria species, and it was included in the discussion the following paragraph to clarify it:

“No eicosapentaenoic acid presence were found for the seaweeds tested in the present work. The presence of this n-3 fatty acid in Gracilaria spp. is inconstant, because it was found in G. gracilis [Francavilla et al., 2013], but it was not detected in G. changii [Chan and Matanjum, 2017] or G. edulis [Satthivel and Pandima Devi, 2015]”

Two of the references cited by you were included in the refences list and was included in the discussion of the current manuscript:

Benjama, O.; Masniyom, P. (2012). Biochemical composition and physicochemical properties of two red seaweeds (Gracilaria fisheri and G. tenuistipitata) from the Pattani bay in Sothern Thailand. Songklanakarin J Sci Technol 2012, 34, 223-230.

Da Costa, E.; Melo, T.; Moreira, A.S.P.; Bernardo, C.; Helguero, L.; Ferreira, I.; Cruz, M.T.; Rego, A.M.; Domingues, P.; Calado, R.; Abreu, M.H.; Domingues, M.R. Valorization of lipids from Gracilaria sp. Through lipidomics and decoding of antiproliferative and anti-inflammatory activity. Mar Drugs 2017, 15, 62.

Because fatty acids according to the comments from other Reviewers were included in a Table with proximate composition, it is difficult to include the percentages in the table. Thus, we included the % of SFA, MUFAs and PUFAS in the text and compared to other works. It was included the paragraph:

“Overall, in G. corticata, SFA accounted 49.4 % of total fatty acids, MUFA accounted a 3.3 % and PUFA accounted a 47.3%, whereas in the case of G. edulis, SFA accounted 43.9 % of total fatty acids, MUFA accounted a 27 % of total fatty acids, and PUFA accounted a 29%. In any case, these profiles were significantly different than 57.5% SFAs, 18.3% MUFAs and 18.4% PUFAs  reported for Gracilaria sp. [da Costa et al., 2017] or the 7.53% SFAs , 38.3% MUFAs and 51.2% PUFAs 18.4% reported for Gracilaria changii [Tan and Matanjum, 2017]”

With respect to the comments about Provide information about replicates in all subsections.

Many thanks for your comments. According to the suggestion from the Reviewer, it was included in all sections of determinations that trials were performed in triplicate.

With respect to the comments about include brief description of the core methods used following AOAC methods, and include the internal standards and how absolute quantification was performed (was based on calibration curves for the different fatty acids?):

Many thanks for your comments. According to the suggestion from the Reviewer, it was included a brief description of proximate analysis methods and was specified in the fatty acid’s determinations that tricosanoic acid was used as internal standard and quantification was performed by comparison to calibrated curves with different fatty acids.

With respect to the comments about the formula employed for the chlorophyll determinations:

Many thanks for your comments. In fact, the formulas employed was taken from a recent article (Satthivel and Pandima Devi, 2015)” that was specified in the revised version of the manuscript.

With respect to the comments about the number of minerals determined

Many thanks for your comments. In fact, only the four cited minerals were determined. It was based on problems of availability of access to equipment in our institution.

With respect to the comments about statistics:

Many thanks for your comments. The comparison between composition of G. corticata and G. edulis was made by means of paired t-test. ANOVA was used for determination of temperature effects in physochemical parameters and were included in the table and text in the revised version of the manuscript.

With respect to the comments about rewriting the conclusions to avoid similarities:

Many thanks for your comments. According the suggestions from the reviewers, conclusion section was rewrite and extended to reduce the similarity index with previous works.

Round 2

Reviewer 1 Report

The revision has been performed according the reviewers’ comments. The paper can be accepted in the present form.

Author Response

Thank you very much for your attention.

Reviewer 2 Report

The authors have improved the manuscript that it is now suitable for publication

Author Response

Thank you very much for your attention.

Reviewer 3 Report

Manuscript was significantly improved and in the current version may be accepted for publication.

Author Response

Thank you very much for your attention.

Reviewer 4 Report

#1 Line 20

 Proximate composition of dried seaweeds revealed a higher content carbohydrates (8.3 ± 1.89 g/100 g), total crude protein (22.844 ± 0.87 g/100 g) and lipid content (7.07 21 ± 0.33 g/100 g) in G. corticata than in G. edulis

The authors decided to represent mean values without standard deviation. But please consider the significant digits please, revising all manuscript and tables.

#2Line 76

Revise the sentence about chlorophyll, that is not correct. It is not the most abundant compound in algae!

In the red color of red algae results from the pigments phycoerythrin and phycocyanin; this masks the other pigments, Chlorophyll a (no Chlorophyll b), beta-carotene and a number of unique xanthophylls.

#3Line 95- Proximate, polysaccharide content and fatty acids profile of both G. corticata and G. edulis in a DW basis  are shown in Table

#4Line 160 Delete as: total fatty acid content, expressed as as

#5line 167 Please carefully revise the designation of fatty acids, that are uncorrected in the table and very confusing in the abstract and results!! Please clarify this part very well.

Alpha-linolenic acid (ALA)

18:3 (n-3)

all-cis-9,12,15-octadecatrienoic acid

Stearidonic acid (SDA)

18:4 (n-3)

all-cis-6,9,12,15,-octadecatetraenoic acid

Linoleic acid (LA)

18:2 (n-6)

all-cis-9,12-octadecadienoic   acid

Gamma-linolenic acid (GLA)

18:3 (n-6)

all-cis-6,9,12-octadecatrienoic   acid

#6Line 185 Table 1. Proximate composition and fatty acids profile of G. corticata and G. edulis. c

Include the units and means in legend

Please divide into two tables, one for biochemical composition another for fatty acids.

How we relate the amount of carbohydrate with polysaccharides? Is it included in the amount 49.64 g/100g of biomass...and the content in carbohydrate is lower. The content is of POS is not presented and the difference is not discussed. Please clarify this part very well.

Attention to significative digits in the table

# 7Line 248

Are you sure is ti chlorophyll b and not c, for example. Chl b is not characteristic of red algae? Or are epiphytes?

# 8Line 370 Correct specie to species

Author Response

With respect to the comments about the need to consider the significant digits:

Many thanks for your comments. According to the comments from the Reviewer, averages values and standard deviation was changed to 2 significant digits, whereas % was changed to a unique significant digit.

With respect to the comments about rewrite the line 76:

Many thanks for your comments. According to the comments from the Reviewer, the phrase was changed from “the predominant” was changed to “an important pigment”

With respect to the comments about line 95:

Many thanks for your comments. According to the comments from the Reviewer, “showed” was changed to “shown”.

With respect to the comments about line 95:

Many thanks for your comments. According to the comments from the Reviewer, “as” was deleted.

With respect to the comments about line designation of fatty acids:

Many thanks for your comments. According to the comments from the Reviewer, designation of fatty acids was checked and corrected in Table, results and abstracts sections.

With respect to the comments about units in Table 1:

Many thanks for your comments. According to the comments from the Reviewer, units was included in the legend of Table 1.

With respect to the comments about split Table 1 in two different tables:

Many thanks for your comments. According to the comments from the Reviewer, Table 1 was divided in 2 different tables.

With respect to the comments about differences in carbohydrates and polysaccharides:

Many thanks for your comments. Please note that carbohydrates and polysaccharides are different concepts, because polysaccharides can include in their structure other compounds that are not carbohydrates. Please note also that in most published articles about seaweeds composition, often polysaccharides are higher than carbohydrates. In example:

Sakthivel, R.; Devi, K.P. Evaluation of physicochemical properties, proximate and nutritional composition of Gracilaria edulis collected from Palk Bay. Food Chem 2015, 174, 68-74.

With respect to the comments about significative digits in the table:

Many thanks for your comments. According to the comments from the Reviewer, significative digits in tables was unified to 2 digits.

With respect to the comments about if we are sure that chlorophyll b is b and not c, for example.

Many thanks for your comments. The chlorophyll a and b determinations in the current manuscript were performed according to the method cited by Saktivel and Pandima Devi (2015) reference. Other works employed close similar method (in example: Chan, P.T.; Matanjun, P., Food Chem 2017, 221, 302-310) and found chlorophyll b content in Gracilaria. Obviously, we only replicated the method described and did not developed it. Thus, we cannot be completely sure of the specificity of the method, but it was performed (and the results are compatible) with than those obtained by other authors.

With respect to the comments about correct specie to species:

Many thanks for your comments. According to the comments from the Reviewer, “specie” was changed to “species”.